# Antarctic Studies Show Lichens to be Excellent Biomonitors of Climate Change

**Leopoldo G. Sancho** [1,*,†], **Ana Pintado** [1,†]  and **T. G. Allan Green** [1,2,†]

1   Departamento de Farmacología, Farmacognosia y Botánica, Facultad de Farmacia, Universidad Complutense, 28040 Madrid, Spain; apintado@ucm.es (A.P.); greentga@waikato.ac.nz (T.G.A.G.)
2   Biological Sciences, Waikato University, Hamilton 3240, New Zealand
*   Correspondence: sancholg@ucm.es
†   These authors contributed equally to this work.

**Abstract:** Lichens have been used as biomonitors for multiple purposes. They are well-known as air pollution indicators around urban and industrial centers. More recently, several attempts have been made to use lichens as monitors of climate change especially in alpine and polar regions. In this paper, we review the value of saxicolous lichens for monitoring environmental changes in Antarctic regions. The pristine Antarctica offers a unique opportunity to study the effects of climate change along a latitudinal gradient that extends between 62° and 87° S. Both lichen species diversity and thallus growth rate seem to show significant correlations to mean annual temperature for gradients across the continent as well as to short time climate oscillation in the Antarctic Peninsula. Competition interactions appear to be small so that individual thalli develop in balance with environmental conditions and, as a result, can indicate the trends in productivity for discrete time intervals over long periods of time.

**Keywords:** Antarctica; biomonitoring; lichens; growth rate; diversity; temperature; precipitation; climate change

## 1. Introduction

Environmental monitoring using lichens has a long and successful history. Although the relationship between lichens and their environment had been noted for many years [1] it was Gilbert [2] who produced the first map linking the concentration of an environmental pollutant, in this case sulphur dioxide, and the abundance and types of lichens. An improved scale by Hawksworth and Rose [3] allows the mean annual $SO_2$ level to be predicted from the lichens present using both species and morphology. The simple rule—the more a lichen stands out from the substrate (tree trunks), the less pollution—seems to hold well [4], although lichen communities can be influenced by other ecological factors, like tree species, forest structure, and microclimatic conditions [5].

In parallel to the diversity measurements, other methodologies are based on the presence/absence of certain lichen species, their morphology, and their abundance. A common alternative is to use the quantity of particular substances taken up over a certain exposure period as a measure of deposition rates, a proxy for atmospheric concentrations. In this case, the lichen acts as an accumulator because they lack the protective cuticles of higher plants and the pollutants are mainly bound to the cell walls. This methodology is both cheap and applicable on a large scale, so both lichens and bryophytes have been used to monitor atmospheric pollutants such as mercury, arsenic, nickel, lead, and other heavy metals [6–8].

In contrast, the use of lichens to monitor climate change has not proved to be so easy. The effects of any changes in broad climatic factors, such as temperature and/or water availability, may be

confounded by other aspects of the environment. In particular, in Europe, there is, and has been, a massive influence by pollutants. In heavily populated and industrial areas, lichen deserts developed in the early 20th century due to high $SO_2$ levels. Levels of this pollutant have fallen dramatically in Europe in past decades, for example, in the United Kingdom from total emissions of 6.49 million tons in 1970 to 180 thousand tons in 2018 [9], and this has led to recolonization by $SO_2$ sensitive species [10,11]. The situation is complicated by a second pollutant, nitrogen as both oxides and ammonia, which remains at high levels in Europe with road traffic being an important recent source. This latter pollutant has led to eutrophication of the habitats, loss of sensitive species, and gains by nitrophytic species [12,13]. In consequence, we have a dynamic situation of previously excluded lichen species returning to areas previously polluted by $SO_2$. Concurrently, there is a loss of $SO_2$-resistant species and gains by species that tolerate or require high nitrogen and also, but often ignored, two groups of potentially new species, one occupying the intermediate period between loss and return, and the other, the completely new habitats created by the changing pollutant impacts. In this situation using lichens to identify change due to climate is difficult, perhaps impossible at the moment and claims to have done so are open to debate [14,15]. A detailed assessment of the situation using historical data (many of the species had a broader distribution in the past) and data from sites where single pollutants occurred is urgently needed. In addition to pollutants that directly impact lichens, they can also be affected by other changes in their environment, in particular, responses of higher plants. Peterson and McCune [16] and Ellis et al. [17] have drawn attention to lichens being affected by changes in forest extent and disturbance whilst Cornelissen et al. [18] have raised the possibility that macrolichens will decline in climatically milder arctic ecosystems where global changes cause vascular plants to increase in abundance. Extensive commercial collecting of lichens can also make it more difficult to detect change [19]. Insarov et al. [20] make the point that use of gradients might be more effective for detecting climate change effects rather than looking just at changes at a single point.

Bearing in mind the comments made above, the most suitable locations to monitor and detect the impact of climate change on lichens would be those with no or very little pollution (both now and in the past), minimal-to-no human impacts, good records of species, and gradients for both lichen and environment. This brings us to Antarctica, which meets all these requirements over almost all the continent and, in the Antarctic Peninsula, has evidence of distributional changes in higher plants in response to climate change [21,22]. The main goals of this paper, therefore, are, for Antarctica: (1) To update information on both lichen growth and diversity; (2) to use meteorological information to link lichen growth and performance to major environmental factors such as temperature and precipitation; (3) to anticipate changes in lichen growth and diversity based on future climate scenarios; and (4) overall, to establish lichens as a major group for monitoring climate change.

## 2. Antarctica

### 2.1. Physical Features

Antarctica is the fifth largest continent with an area of 14,000,000 km$^2$ and is almost circular in form, approximately centered on the South Pole (Figure 1). It is the world's highest continent with an average height of 2500 m, mainly due to most of the continent being an ice sheet with thickness reaching over 4 km and containing 80% of the world's freshwater and almost 90% of the ice. Low-altitude areas are normally close to, or at, the coast. The area composed of exposed rock is very small, at 21,745 km$^2$ or 0.18% of the continent [23]. Much of this exposed rock occurs in the Antarctic Peninsula, which is 1300 km long, stretching from 63° S to 73° S, and in the Transantarctic Mountains that stretch from northern Victoria Land, around 71° S, along the coast of the Ross Sea and the Ross Ice Shelf to just over 87° S (Mount Howe) (Figure 1).

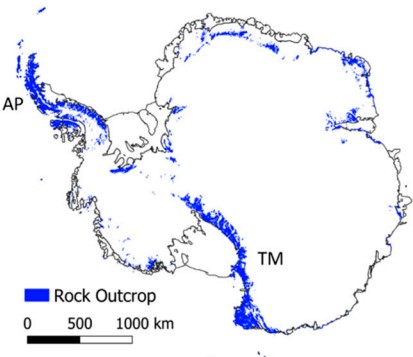

**Figure 1.** Distribution of ice-free land in Antarctica. The two areas with the largest proportion of ice-free land on the continent are named the Transantarctic Mountains (TM) and the Antarctic Peninsula (AP). Figure by A. Burton-Johnson using the Landsat 8-derived rock outcrop data of Burton-Johnson et al. (2016) [21].

*2.2. Human Influence*

Antarctica has no permanent human inhabitants but more than 80 research stations exist and about half of them are staffed for the full year. Human influence on the majority of the ice-free areas is usually minimal to nothing with the main impacts being in the north of the Antarctic Peninsula and on the adjacent islands. Although minute traces of man-made chemicals used in other parts of the world can be detected in Antarctica, pollution levels are very low except, occasionally, in the near vicinity of bases [24].

*2.3. Vegetation*

Only two species of vascular plant are found in Antarctica, *Deschampsia antarctica* and *Colobanthus quitensis*, only in the Antarctic Peninsula and then mainly in the more northerly sites where they can be locally dominant [21]. The terrestrial vegetation is dominated by lichens and bryophytes. Both cryptogams are poikilohydric and this appears to be the underlying reason for their dominance in Antarctica with homoiohydric plants confined to the more northerly sites [22]. There are around 400 species of lichens which preferentially occupy drier sites or rock surfaces throughout Antarctica. Bryophytes, total about 115 species, are most abundant in wet habitats from which lichens are excluded [25] and are divided into two main groups, mosses and liverworts. Liverworts do not extend as far south as mosses and lichens and reach their southern limit at Botany Bay, 77° S.

**3. Gradients**

Monitoring the location of species along an environmental gradient is one method to look at the effects of climate change [26]. It is based on the assumption that, for species to survive, they must shift their distributions to track preferred conditions [27,28]. Gradients will almost certainly show the direction in which the response to climate change will occur, so gradients represent a space-for-time situation [29]. However, actual speed of change is likely to be slow as various factors such as surface composition, rates of dispersal, and degree of plasticity will influence how rapidly such changes can occur, and almost certainly introduce some short-term blurring of the process. Environmental gradients are more useful when there is little or no interference from other factors that cause vegetation change and increase in temperature and changes in precipitation are clear. In view of its size, it is not surprising that there are major environmental gradients in the latter factors across Antarctica.

*3.1. Environmental Gradients Across Antarctica*

The most obvious gradient is in mean annual temperature which increases at approximately 0.7 °C per degree latitude from –19.9 °C at Scott Base, Ross Island, 77.85° S, to –2.3 °C at Bellingshausen, 62.20° S, at the north of the Antarctic Peninsula [30]. The gradient is constructed from data from

bases that are all located on the coast and the relationship is highly significant ($P = 0.0058$, $r^2 = 0.504$). There is also a large increase in precipitation (mm rain equivalent) from only about 50 mm in the Dry Valleys (77.8° S), through to 225 mm at Cape Hallett (72.0° S), 800 mm at Livingston Island (62.6° S), and 400 mm at the sub-Antarctic Signy Island (60.7° S) (Figure 2). Information about precipitation is much more limited than for temperature because of the many difficulties in its measurement, as it can fall as snow or rain depending on location and season. There are also steep altitudinal gradients present at almost all points along the coast but detailed information is not available for these.

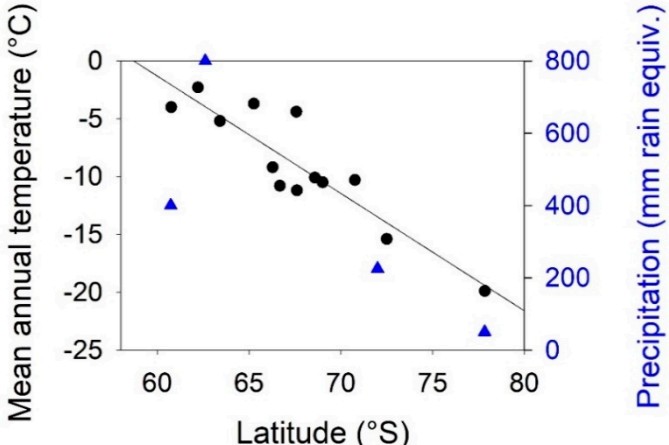

**Figure 2.** Mean annual temperature (•, °C) and annual precipitation (▲, mm rain equivalent) versus latitude across Antarctica. Temperature data are from 13 Antarctic bases whilst those for precipitation are from four sites (Dry Valleys, Cape Hallett, Livingston Island, and Signy Island).

*3.2. Lichen and Bryophyte Gradients Across Antarctica*

Both lichens and bryophytes show strong responses that correlate with the environmental gradients across Antarctica.

Species number: Both groups show a decrease in total number of species with increase in latitude across Antarctica (Figure 3) declining from around 350 lichen species in the north of the Antarctic Peninsula, to about 120 in the south, then to between 30 to 40 at continental sites as far south as 84.4° S (Mt. Kyffin) and still around 12 species at the extreme latitude of 87° S (unpublished results [30]). The equivalent numbers for mosses are 100–115, 40–50, 2–8, and 2, and for liverworts 27, 2, 0, and 0 [31,32]. Liverworts have their southern limit at Botany Bay, 77° S [33]. There is also a highly significant, strong linear relationship between number of species of the three groups of photosynthetic organisms (% maximum number per ° latitude) and mean annual temperature [30]. There is a decline in species numbers of 9.8%, 9.2%, and 10.0% per °C fall in annual mean temperature for lichens, mosses, and hepatics, respectively. According to the regression lines, zero species will occur at –13.7, –14.4, and –10.9 °C for lichens, mosses, and hepatics, respectively (Figure 3) which are equivalent to the latitudes 72°12′ S, 72°54′ S, and 69°25′ S (approximately the northern limit of the Ross Sea). In reality, all groups occur much further south and Green et al. [30] suggest that the zero occurrence on the regression marks a shift from control by macroclimate to the north to control by microclimate to the south. This shift appears to be related to water availability and marks the point at which incident precipitation is no longer sufficient to maintain growth and the vegetation becomes confined to sites in which water is "concentrated" i.e., by being locally abundant due to melt from snow banks and landscape topography. Within the Ross Sea, therefore, lichen numbers (for which the data are better than for other groups) are dictated by local environmental conditions rather than global climate.

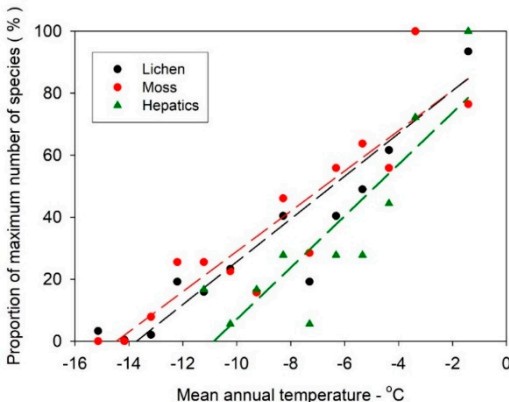

**Figure 3.** Number of species of lichens, mosses, and hepatics as a percentage of their maximum number (usually close to warmest place) plotted against mean annual temperature. The fitted lines are linear regressions all *P* < 0.0001, r$^2$ are 0.83, 0.84, and 0.75, respectively. Data for Antarctic Peninsula and Ross Sea.

Lichen growth rates: Studies on lichen growth rates are rare in Antarctica but data sets exist for four sites that span the continent, Signy Island, 60°43′ S, Livingston Island, 62°39′ S, Cape Hallett, 72°19′ S, and Dry Valleys, 77°40′ S [34]. At all sites, the measured lichen thalli are separate from each other so that changes in growth rate are driven by habitat environment rather than biotic interaction. This situation normally arises because the lichens are colonizing recently exposed surfaces often by glacial retreat. Growth rates (radial elongation) are extremely different at these sites ranging from close to the fastest for saxicolous crustose lichens at the two northern sites (0.47–0.50 mm yr$^{-1}$ radial elongation) to the slowest yet reported of <0.01 mm yr$^{-1}$ in the Dry Valleys (Table 1). There is a significant linear relationship (*P* = 0.252, r$^2$ = 0.925) between log$_{10}$ growth rates and mean annual temperature for these sites (Figure 4). Growth rates differ by almost two orders of magnitude between −1.3 °C and −20.1 °C. It is a good question as to exactly why this relationship occurs as, for most of the year, in winter, and especially in continental sites, the lichens are dormant. However, the strong relationship remains when the growth rate and number of lichen species is plotted against mean temperature of the warmest month (Figure 4a,b). Both log$_{10}$ lichen growth rate and log$_{10}$ number of lichen species show almost identical relationships with annual precipitation (mm rain equivalent) but, in this case, the responses are non-linear and shows signs of saturation at annual precipitation greater than 400 mm (Figure 4c; [34]). The similarity of the relationships to temperature for growth rates and species numbers strongly suggests that the underlying physiological mechanisms are identical, or very similar, in both cases, and are likely to represent changes in carbon allocation strategies in response to changes in habitat stress and total carbon gain by the lichens [35].

**Table 1.** Radial growth rate of Buellia (* *Buellia frigida*; ** *Buellia latemarginata*) species and main climatic parameters from Antarctic localities (Sancho et al. 2007, modified).

| Location | Growth Rate Radial (mm y$^{-1}$) | Warmest Month (Mean T °C) | Coldest Month (Mean T °C) | Annual Mean (T °C) | Precipitation (mm Rain Equivalent) |
|---|---|---|---|---|---|
| Dry Valleys (*) | 0.01 | −4.8 | −30.5 | −20.0 | 50 |
| Cape Hallett (*) | 0.07 | −1.4 | −26.4 | −15.0 | 120 |
| Signy Island (**) | 0.25 | 1.3 | −9.0 | −3.3 | 400 |
| Livingston Island (**) | 0.44 | 1.3 | −7.0 | −1.5 | 800 |

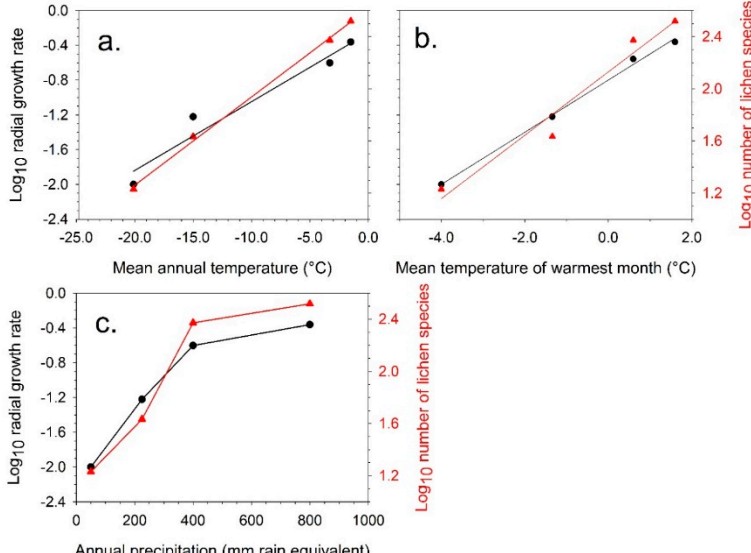

**Figure 4.** Relationship of $\log_{10}$ lichen radial growth rate (●) and $\log_{10}$ number of lichen species (▲) with (**a**) mean annual temperature (°C), (**b**) mean temperature of warmest month, and (**c**) annual rainfall (mm rain equivalent). Significance of regressions in (**a**) are $P = 0.0252$, $r^2 = 0.925$ and $P = 0.0009$ and $r^2 = 0.997$, and in (**b**) $P = 0.0019$, $r^2 = 0.994$ and $P = 0.0198$ and $r^2 = 0.941$, for lichen radial growth rates and number of lichen species, respectively.

Photobiont type: Green et al. [30] showed that lichens with cyanobacterial primary photobionts are effectively excluded from continental Antarctica and barely extend further than the southernmost part of the Antarctic Peninsula. There are good physiological reasons for this, as both net photosynthesis and nitrogen fixation do not seem to occur below 0 °C in cyanobacteria. The present distribution suggests expansion from refugia within, or outside Antarctica, such as offshore islands [22] in the Antarctic Peninsula region after the last glacial maximum. Further expansion depends on success in dispersal as the ice-free areas near the base of the peninsula tend to be widely separated from one another (Figure 1).

## 4. Short-Term Effects of Climate on Lichen Growth

### 4.1. Short-Term Growth Rate Studies

Gradient studies look mainly at long-term changes and, although they are good indicators of direction of change, the actual rate can be slow because of dispersal and habitat availability limitations. At the northern end of the lichen growth rate gradient in Antarctica (the north part of the Antarctic Peninsula) response to precipitation seems to tend toward saturation so that further increases in precipitation will have a lesser effect. In this location, one might expect the influence of temperature on growth rate to become more important and, because lichen growth rates are high, changes to become more obvious allowing short-term responses to be detected. In addition, temperature changes over recent past decades have been large in the north of the Antarctic Peninsula. First, a steady and rapid warming from the 1950s to 1998 and, since then, an equally dramatic decline [36]. Long-term lichenometric studies in front of a receding glacier on Livingston Island have shown that the effects of such climate changes can be rapid and massive [37]. A total of six lichen species were measured in 1991, 2002, and 2015; one species was fruticose (*Usnea antarctica*) and the remaining five were crustose. The mean temperature of the three summer months (December to February, inclusive) rose from 1.06 °C in 1991 to 1.48 °C in 2002 and then declined to 0.90 °C in 2015, with 0.58 °C being the total temperature span. *Usnea antarctica* showed an almost perfect linear correlation between growth rate and mean summer temperature of 0.74 mm yr$^{-1}$ degree$^{-1}$ ($r^2 = 0.998$, $P = 0.029$). The remaining species showed a wide range of different species–specific responses ranging from almost no change

for *Buellia latemarginata* to slight increases in growth rate in the warming between 1990 and 2002 for the other species. However, these remaining species all showed dramatic declines in apparent growth rate in the cooling period between 2002 and 2015. *Caloplaca sublobulata* and *Acarospora macrocyclos* even showed negative growth, which is not possible for crustose lichen species. This result is a consequence of these species having suffered, to some degree, a loss of larger thalli as a result of being covered by snow for longer periods. Snow kill appears to be an important process in Antarctica, as well as in alpine and arctic environments [38,39] and leads to rock surfaces being made available for new colonisation. The same process was suggested for the establishment of the crustose lichen *Buellia frigida* in the Dry Valleys, albeit at around 1000 and 4000 years ago [40].

## 4.2. Predicted Effect of Increasing Temperature for Different Antarctic Lichen Species

There seems to be a general consensus that there will be a negative influence by climate warming on lichen cover, both in Arctic-alpine tundra [41,42] and in Antarctica [43]. However, these experiments showed that this was not a direct effect of the temperature being warmer but, rather, the consequence of an increase in vascular vegetation or a deleterious effect of anomalous snow cover trapped inside the artificial warming facilities [43,44]. Our long-term observational site on Livingston Island can help clarify the real response of saxicolous lichens to environmental changes. The locality consists of a young moraine close to a glacier front and is made up of rock boulders so that competition with vascular plants, and even with mosses, can be considered as being negligible. Over the past 25 years of observations, two very contrasting climatic periods have occurred in this Antarctic region [36]. Two decades of strong warming in the 1980s and 1990s, followed by 15 years of cooling in the present century (change year being 1998) produced very clear species-specific responses [37]. When changes are calculated on the basis of a 0.5 °C warming or cooling (Figure 5) we can see how *Acarospora macrocyclos* and *Bellemerea* sp. slightly increase their annual growth rate (11% and 16%, respectively) by 0.5 °C of warming, but are dramatically, and negatively, affected by 0.5 °C of cooling with their growth rates reduced by 137% and 60%, respectively. Something similar happen to *Caloplaca sublobulata* with the species showing a small, negative effect of 0.5 °C warming (15% decrease) but a drastic reduction of its growth rate (115%) by cooling. *Usnea antarctica* and *Rhizocarpon geographicum* show a very similar increase (49%, 28%) and decrease (65%, 25%) for 0.5 °C of warming or cooling, respectively. Interestingly, *Buellia latemarginata* does not show any significant change during these climatic periods.

The obvious conclusion is that, once competition with other more productive organisms, mainly vascular plants, has been excluded, most lichen species of the Antarctic tundra react positively to a warming climate and negatively to cooling. This is not surprising for two reasons. The optimum temperature for photosynthesis of the investigated Antarctic lichens and mosses is typically between 10 and 17 °C under moderate to high radiation [45–48], just a little lower than values reported for the Antarctic vascular plants [49]. Air temperature, even in the warmest Antarctic regions, is far below this value, suggesting that photosynthesis in Antarctic vegetation is limited by low temperatures. In addition, there is a large increase in active time across Antarctica and this delivers a much larger gain than from an increased net photosynthetic rate [50]. Together, the response of the two processes to a warming climatic scenario will produce an increase in the primary productivity of the components of the Antarctic tundra. Knowledge about the influence of this apparently positive effect on biomass and, even more important, on functional, phylogenetic, and species diversity will be crucial for consistent predictions and models.

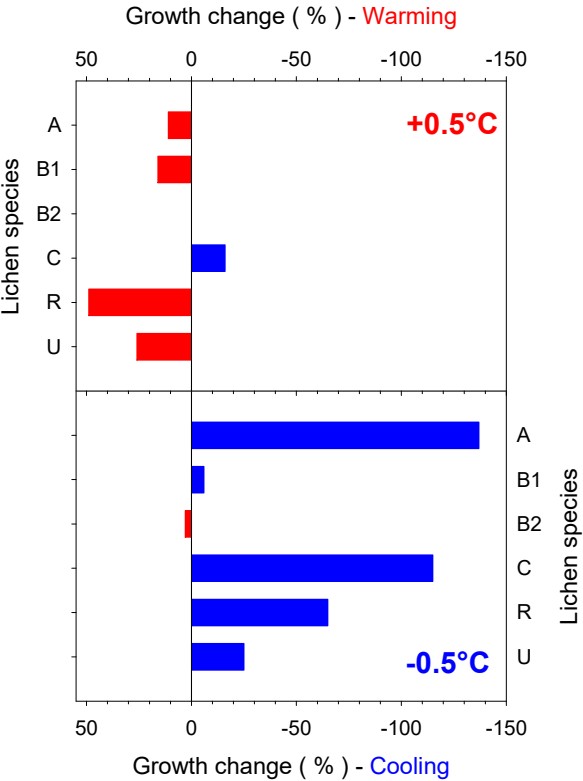

**Figure 5.** Changes in thallus growth rate observed for six lichen species on Livingston Island. Upper panel—warming of 0.5 °C, lower panel—cooling of 0.5 °C; bar colour: Red, increased growth rate, blue, decreased growth rate. Species: A—*Acarospora macrocyclos*; B1—*Bellmerea* sp.; B2—*Buellia latemarginata*; C—*Caloplaca sublobulata*; R—*Rhizocarpon geographicum*; U—*Usnea antarctica*.

## 5. Discussion

Antarctica offers a unique opportunity to utilise a single functional group, lichens, to monitor climate change. Several features of the continent underpin this possibility:

1. Antarctica is unique in having no resident human population and human impacts on terrestrial vegetation are zero-to-minimal except in close proximity to bases.

2. Antarctica has the lowest pollution levels in the world and well below levels likely to influence terrestrial vegetation.

3. Suitable habitats (rock surfaces) span almost the entire latitudinal range across the continent.

4. Antarctica is large and has environmental gradients, in particular, temperature and precipitation, that span almost the complete range of growth rates of crustose lichens from lowest known to amongst the fastest.

5. Antarctica has no higher plants over most of its surface, in particular, no woody plants, that would compromise lichen habitats.

The terrestrial macrovegetation is composed almost entirely of lichens and bryophytes. Lichens, which are mainly saxicolous, offer, by far, the better opportunity to monitor climate change because they are likely to be more tightly coupled to the climate whereas the bryophytes are mainly confined to very wet areas, with temperature moderated by the water flowing around them, and have a smaller number of species. However, bryophytes do preserve growth-rate information in internode length along their stems and so offer a very useful local information source especially about water availability [51]. Hepatics also represent a special case as they have a southern limit within Antarctica and this offers some possibility to track change through range extension or contraction.

Lichens offer several excellent means to track climate effects at several scales, geographic and temporal.

1. They have a substantial number of species in Antarctica (around 400) and the number present at any particular site shows a tight relationship with mean temperature with a change rate of around 10 species per °C mean annual temperature. However, such a change is a long-term process due to potential limitations from habitat availability and dispersal ability. The latter process not being that well understood at present but departures from the main regression relationship might well indicate that other processes, like dispersal, are having an effect.

2. One functional group of lichens, those with cyanobacterial primary photobionts, have a southern limit within Antarctica and any warming is likely to cause further range extension. The likely physiological driver for this limit is known (limits to photosynthesis and nitrogen fixation below 0 °C) but not well understood.

3. Short-term (sub-decadal) changes in growth rates can be measured at the north of the Antarctic Peninsula and these allow changes in temperature to be rapidly detected and tracked. However, responses are clearly species-specific and also confounded by abrupt effects such as snow kill. The latter, however, does produce new surfaces for colonisation.

*Possible Drivers of Lichen Diversity and Growth*

Across polar areas to temperate regions, at geographical scales, there is substantial evidence for a broadly positive monotonic relationship between species richness and energy availability, due to an increase in net productivity [52]. The same is found with altitudinal gradients [53,54]. There is good evidence that productivity is higher at lower latitudes in Antarctica. The increase does not appear to be due to increased absolute net photosynthetic rates which appear to be lower in Antarctica than in temperate areas [30,55] and constant across the continent, although data are few [50]. The most likely cause appears to be longer activity periods at warmer sites. Monitoring of active periods using non-contact chlorophyll fluorescence shows much higher activity (% of time active) in the Antarctic Peninsula than at Botany Bay and, particularly, in the Dry Valleys [56]. This increase is present even when only summer months are considered but there are major differences between epilithic lichens (low activity) and flush bryophytes, which show very high activity whilst wet [47,57,58]. When one considers that both lichens and bryophytes are poikilohydric and need liquid water to fully hydrate, then the link with higher temperatures, i.e., above 0 °C the melt temperature of water, is not surprising. In Antarctica there is also a strong correlation between annual precipitation and mean annual temperature so that it is difficult to disentangle these drivers. However, in the north of the Antarctic Peninsula, the rate of response in both growth rate and species numbers to increased precipitation declines whereas the linear response to temperature remains (Figure 4).

An increase in niches is also an important contributor to the gain in diversity [30]. Antarctica was divided into two environmental zones, the macroenvironmental zone, north of about 72° S in which vegetation was increasingly influenced by global climate, and the microenvironmental zone to the south of 72° S in which vegetation is influenced by the local occurrence of water [30]. The zone boundary is where the regression of species number to mean annual temperature reaches zero. In the microenvironmental zone, the lichens are effectively landscape controlled in the sense that they can only colonize and grow where water is "concentrated", for example by melt from snow patches or glaciers. North of the boundary, the precipitation is sufficient for lichens to start to colonise surfaces not receiving this "concentrated" water, i.e., they start to move out onto rock surfaces. This not only allows greater cover, but also greater diversity as new habitats select new species. There is also a probable shift towards using recently fixed carbon for growth rather than stress tolerance [35]. It appears that the major changes along the gradients do not represent adaptation to new conditions by existing lichens but are driven mainly by the arrival of new species with different adaptation. A similar situation (directional-replacement) is reported for the Arctic for vegetation succession in Ellesmere Island [59].

Physiological lifestyle strategy: The number of lichen species increases rapidly with rising mean annual temperature and precipitation across Antarctica. Change is, therefore, predominantly by new species addition although adaptation by species that span a large latitudinal range must also be considered. Colesie et al. [60] showed, for lichen-dominated soil crusts, that carbon allocation strategies change, so that newly fixed carbon is predominantly allocated to stress protection in the Antarctic and to growth at a temperate site. This represents a lifestyle gradient but, at present, we lack any real data to show how species differ or agree at any particular site. Although there has been a focus on adaptation by a species as a means to adjust to climate change, Bjorkman et al. [61] make the point that spatial trait–environment relationships are driven largely by species turnover, suggesting that modelling efforts must account for rates of species immigration when predicting the speed of future functional shifts [59].

## 6. Future Possibilities

Lichens seem to offer possibilities for future research that could contribute considerably to our understanding of drivers of climate change responses. Although net photosynthetic rates are apparently both low [55] and stable [48] within Antarctica, there is evidence that allocation patterns of recently fixed carbon change from stress toleration to growth. This is a topic that could be developed with single species that span the continent (*Umbilicaria decussata*) or by comparing several species. Results from temperate areas on the soil crust lichen *Psora decipiens* have revealed considerable morphological flexibility in this widespread lichen that appears to be strongly linked to water budget [35,62]. This flexibility also includes photobiont switching which is becoming an important topic in lichen ecophysiology. However, it may not be so important in Antarctica where photobiont variability might be constrained by the harsher habitat conditions [63,64] Again, the gradients in Antarctica offer major possibilities for this type of research.

Molecular studies on lichens are still in their infancy in Antarctica, but results suggest that this could also be an area that will contribute to our understanding of climate drivers. Recent studies on bryophytes [65], lichens [63,66], and survival and evolution in collembolids [67] strongly suggest that an interesting evolutionary story is present within the genomes of these terrestrial organisms in Antarctica. Genome variability is often spatially determined, indicating that it is preserving information about the past history of the species. Genomes are not visible, so, in order to prevent contamination of these patterns, there is an important need for so-called "internal quarantine", that is the prevention of the mixing of genomes by accidental movement of organisms between sites within Antarctica.

**Author Contributions:** All three authors contributed equally to the paper.

**Funding:** This research was funded by Ministerio de Ciencia, Innovación y Universidades CTM-2015-64728-C2-1.

**Acknowledgments:** Burton-Johnson, A, (British Antarctic Survey) is thanked for assistance with Figure 1.

**Conflicts of Interest:** The authors declare no conflict of interest. The funders had no role in the design of the study; in the collection, analyses, or interpretation of data; in the writing of the manuscript, or in the decision to publish the results.

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
