# Peer review of "Antarctic Studies Show Lichens to be Excellent Biomonitors of Climate Change"

_diversity, doi:10.3390/d11030042_

Round 1
Reviewer 1 Report
The manuscript (MS) is about lichens of Antarctica as biomonitors of climate change. The importance of this polar region is undoubted so any study that carried out there is interesting. However, and regrettably, I believe that the manuscript cannot be accepted for publication in Diversity. The first thing that the authors should provide is a Material and Methods of the study. I don’t know if the present results are found by the authors or they come from bibliography. And a table with all the lichens found in the study is not provided either. This is very confusing when you read the manuscript. The authors must to rewrite the paper and provide all the information and reorganizing the sections again. It’s really an interesting topic but the manuscript as it is, it's incomprehensible with an important lack of information and wrong information:
- The Keywords are almost a repetition of the title.
- The Introduction is poor. There is almost no mention of studies carried out in Antarctica with lichens as a biomonitors. Likewise, there are a lot of papers about the use of lichens to monitor climate change so I can only disagree with the authors (line 44). The assertion that mosses do not have a cuticle is not true at all, it is usually very fine, but they have.
- After the Introduction there are a lot of sections but what are they? Because in some of them there are the explanations of the results. Some results that I don't know how they came about. How was the sampling done? Did the authors do it? I assume that the lichens were on rocks? or also in the soil? A list with all the lichens found?? How am I supposed to review the manuscript if I don't know how the data was obtained?? That information should go in the MS and must be reorganized in a logical way if not it is impossible to revise.
- Section Species Number: the line 147 explains Figure 3 and the “zero species” for lichens, mosses and hepatics but if you see the Figure…. THERE ARE NO EXIST ZERO VALUES FOR LICHENS AND HEPATICS! These values should be in the zero line… if not… anyone who sees that figure will not understand that there are zero species.
- Discussion and Conclusions: There is a conclusion in page 11 before the Discussion…. How????? And all sections before the discussion have more discussion than the discussion itself. The whole MS it is a little chaos.
Author Response
Review 1:
The manuscript (MS) is about lichens of Antarctica as biomonitors of climate change. The importance of this polar region is undoubted so any study that carried out there is interesting. However, and regrettably, I believe that the manuscript cannot be accepted for publication in Diversity. The first thing that the authors should provide is a Material and Methods of the study. I don’t know if the present results are found by the authors or they come from bibliography. And a table with all the lichens found in the study is not provided either. This is very confusing when you read the manuscript. The authors must to rewrite the paper and provide all the information and reorganizing the sections again. It’s really an interesting topic but the manuscript as it is, it's incomprehensible with an important lack of information and wrong information:
The fact that this manuscript is a review and not a paper is probably the origin of a misunderstanding that could explain part of the concerns raised by referee 1. This manuscript lack of a chapter on Material and Methods because it is a review and the vast majority of the data used for this study have been already published in multiple papers (all of them cited in the text). This has now been more clearly stated so that the readers can go to original papers if they wish to see the methods used.
- The Keywords are almost a repetition of the title.
This is normal and often requested by the journals. We are willing to change them if this is the editor decision.
- The Introduction is poor. There is almost no mention of studies carried out in Antarctica with lichens as a biomonitors. Likewise, there are a lot of papers about the use of lichens to monitor climate change so I can only disagree with the authors (line 44). The assertion that mosses do not have a cuticle is not true at all, it is usually very fine, but they have.
Yes, the referee is right. We meant “protective cuticle”. Anyway to avoid confusion we have removed “bryophytes” in this sentence. We have also altered the title to indicate that this is a review supporting the use of lichens as biomonitors for climate change. There are few studies specifically on this topic because of the long periods of time needed in such a cold environment to get changes.
- After the Introduction there are a lot of sections but what are they? Because in some of them there are the explanations of the results. Some results that I don't know how they came about. How was the sampling done? Did the authors do it? I assume that the lichens were on rocks? or also in the soil? A list with all the lichens found?? How am I supposed to review the manuscript if I don't know how the data was obtained?? That information should go in the MS and must be reorganized in a logical way if not it is impossible to revise.
Inferences are specific to lichens growing on rocks. This has been included in Abstract. Climate responses of soil or epiphytic lichens may differ (also answer referee 2).
- Section Species Number: the line 147 explains Figure 3 and the “zero species” for lichens, mosses and hepatics but if you see the Figure…. THERE ARE NO EXIST ZERO VALUES FOR LICHENS AND HEPATICS! These values should be in the zero line… if not… anyone who sees that figure will not understand that there are zero species.
The reason is that zero values do not exist. In other words, that means that the regression line reaches zero species, not that there are zero species data points. Actually, this is one of the most important points in the discussion: The zero point estimated after the regression line does not agree with the reality and we try to explain this apparent contradiction thought the microclimatic conditions.
- Discussion and Conclusions: There is a conclusion in page 11 before the Discussion…. How????? And all sections before the discussion have more discussion than the discussion itself. The whole MS it is a little chaos.
This is a review. Please, see the answer to the first question.

Reviewer 2 Report
The manuscript title “Lichens as biomonitors of climate change in Antarctica” has interest and it can be published in Diversity with minor modifications.
The paper is an original review about the use of lichens as indicators of climate change. It is noteworthy that the authors point out the problem existing in many areas of the world to test the effect of climate change on lichens because another antropogenic factors may be also acting. So, they propose the Antarctica as a perfect area to test this aspect because Antarctica is a pristine territory.
Some comments are included in the text.

Author Response
Review 2:
The manuscript title “Lichens as biomonitors of climate change in Antarctica” has interest and it can be published in Diversity with minor modifications.
The paper is an original review about the use of lichens as indicators of climate change. It is noteworthy that the authors point out the problem existing in many areas of the world to test the effect of climate change on lichens because another antropogenic factors may be also acting. So, they propose the Antarctica as a perfect area to test this aspect because Antarctica is a pristine territory.
Some comments are included in the text.
All comments and suggestions made by reviewer 2 have been considered and the manuscript was changed accordingly.

Reviewer 3 Report
Overall:
- In this literature survey, the authors unite growth performance and species-count metrics, measured with both gradient and local observational methods, in a joint approach which is very promising for understanding climate change responses.
- The biggest issue is that many claims/predictions are inconclusive or unsubstantiated: that is, much of the presented evidence is necessary but not sufficient to exclude alternative outcomes. Solutions may be to provide more evidence (where available), or else make statements conditional rather than declarative.
- May be useful to simply state in Abstract that this is a literature survey which does not introduce new data/evidence.
- May be useful to clearly state that inferences are specific to lichens growing on rock and/or soil (depending on what you mean). Climate responses of epiphytes or elsewhere likely differ widely.
Specific comments by line number: ---------------------------------------------------
10 and throughout: Should probably specifically refer to "saxicolous" or "terrestrial" lichens depending on what you mean, since responses not likely to be uniform among all substrates/lifestyles.
15: "show a clear response" is poor wording that implies causation. All you can really assert is that diversity and growth rate were "correlated" with some temperature values along the gradient.
18: "in balance with environment" implies an equilibrium that you later contradict when implying that disperal limitations contribute to non-equilibrium.
51-54:This sentence is awkward and hard to read.
61: The van Herk/Aptroot papers explicitly accounted for responses independent of NH3 and SO2 alterations, so what's up for debate?
72: Also requires appreciable exposure to climate anomalies! Can you provide/cite non-lichen evidence of this for Antarctica?
79: "to predict changes in lichen": True prediction seems too ambitious given your subsequent evidence -- you may guess/expect certain outcomes, but your evidence as given does not preclude nor exclude alternatives.
95-99: This sentence is awkward and hard to read.
104: possible circular argument: Why exactly should poikilohydry relate to abundance? Could explain that non-frozen water availability constrains growth.
110: In addition to growth and richness, this gradients section could benefit from mention of "local abundance" (frequency/cover/biomass) as a monitoring variable of interest; just a thought.
114: "Gradients will... show in which direction change will happen": Not necessarily true: e.g., warming may be greater (and faster) at a currently "cool" site than at a "warm" site; or vice-versa! Your statement doesn't exclude alternative outcomes.
137: Is species number estimated at a single site? in a region? in a latitudinal band? Data collection methods are never really explained.
152-154, and 154-156: No evidence presented for these claims. Did someone simultaneously measure microsite water availability in relation to macroclimate? Does not appear that Green et al. (2011) explicitly did so.
172-173: I'd accept that carbon-balance physiological mechanisms could influence growth rates. But less clear is how physiology need necessarily influence species numbers (why can't one site have 50 poorly-performing species, and another have 5 excellently-performing species? or vice-versa?). There's no evidence to rule out either outcome.
178: Your glacial refugia hypothesis may be consistent with the present distribution, but further evidence would be needed to exclude alternative hypotheses. How would you rule out some other alternative (like post-LGM arrival from other continents and habitat filtering totally independent of glaciation)? What about population genetic structure, or a mechanistic dispersal model?
186-188: This sentence is awkward and hard to read.
199: Usually we report r2 or R2 and a p-value when making a statement about correlation.
220-221: But surely competition is not negligible among saxicolous and terricolous lichen thalli... See work by Armstrong, Maestre, Pastore, ...
222-227: Not sure why this reiterates lines 192-199, from references [33,34].
242: "optimum temperature for photosynthesis" finally suggests how climate may impact the physiological basis of your observed growth changes. Perhaps elevate this sooner or more frequently throughout?
247: Your statement neglects that ecosystem net primary productivity necessarily includes respiration losses (from cellular respiration, or microbial decomposition) that might outpace photosynthetic gains -- what is your expectation for ecosystem respiration?
249: You present no evidence that biomass was measured or estimated -- perhaps you assumed that radial growth must yield more biomass? But that's not necessarily true if larger thalli are also thinner.
252-264: This is a very nice concise summary justifying your program.
266: Might be useful to mention that saxicolous lichens are somewhat decoupled from soil moisture (if indeed you continue to refer only to saxicolous), and so may be useful indicators of atmospheric process.
277-278: So departures from a pure-climate model, therefore, could indicate the magnitude of dispersal limitation (i.e., how far from equilibrium).
291: Not sure why this reiterates lines 172-173, which are likewise unsubstantiated (contradictory outcomes equally plausible/possible).
312-320: I don't doubt this may be true, but no evidence is given! This is probably consistent with the authors' detailed and extensive natural history observations at great effort, but the claim can only be exclusively supported with both microclimate/macroclimate measurements to the exclusion of alternative explanations.
325-33: This is a concise, clear paragraph showing how physiology may scale to landscapes - nice!
347-353: Nice that you mention molecular tools, but might also help to mention that molecular studies can reveal photobiont switching which may expand/constrain climatic tolerances of the symbiosis.
352: What is internal quarantine? Is this a biological concept of gene flow, or a social principle to prevent novel introductions?
### end ###
Author Response
Review 3:
Overall:
- In this literature survey, the authors unite growth performance and species-count metrics, measured with both gradient and local observational methods, in a joint approach which is very promising for understanding climate change responses.
- The biggest issue is that many claims/predictions are inconclusive or unsubstantiated: that is, much of the presented evidence is necessary but not sufficient to exclude alternative outcomes. Solutions may be to provide more evidence (where available), or else make statements conditional rather than declarative.
- May be useful to simply state in Abstract that this is a literature survey which does not introduce new data/evidence.
- May be useful to clearly state that inferences are specific to lichens growing on rock and/or soil (depending on what you mean). Climate responses of epiphytes or elsewhere likely differ widely.
It is now made clear in specific sections of the paper that the lichens studied are saxicolous; when total numbers are considered these will include lichens on soil and epiphytic on mosses.
Specific comments by line number: ---------------------------------------------------
10 and throughout: Should probably specifically refer to "saxicolous" or "terrestrial" lichens depending on what you mean, since responses not likely to be uniform among all substrates/lifestyles.
We have tried to do this as appropriate, the comment is accepted.
15: "show a clear response" is poor wording that implies causation. All you can really assert is that diversity and growth rate were "correlated" with some temperature values along the gradient.
Text modified to make it clear that there were significant correlations with temperature
18: "in balance with environment" implies an equilibrium that you later contradict when implying that disperal limitations contribute to non-equilibrium.
Text altered to clarify that this refers to individual thalli, which are the basis of the growth measurements.
51-54: This sentence is awkward and hard to read.
The sentence has been rewritten and divided into two sentences.
61: The van Herk/Aptroot papers explicitly accounted for responses independent of NH3 and SO2 alterations, so what's up for debate?
We have modified the sentence to make it less critical but the results in this paper are certainly up for debate.
This paper makes use of observational data only from the Netherlands. Larger data bases exist elsewhere and these tend to show that many of the cited species had much more extensive distributions in the past and are now recovering rather than spreading to new sites. Also some species are cited as nitrophiles in other papers.
For example, the paper makes the statement: One example is Flavoparmelia soredians, a drought-resistant, warm-temperate species which until recently had its northernmost limit in southern England (Seaward and Coppins, 2004).; however, the cited paper is about hypereutrophication, which is certainly a phenomenon in the Netherlands. Bates et al. 2007 state about the Seaward paper: inducing the nitrophytic species that are normally associated with eutrophication or hyper-trophication (Seaward and Coppins, 2004)
72: Also requires appreciable exposure to climate anomalies! Can you provide/cite non-lichen evidence of this for Antarctica?
Added a comment that there is possible distributional changes in the higher plants in the Antarctic Peninsula.
79: "to predict changes in lichen": True prediction seems too ambitious given your subsequent evidence -- you may guess/expect certain outcomes, but your evidence as given does not preclude nor exclude alternatives.
We feel the word predict is reasonable in this paper, it is certainly often used elsewhere and we use the significant correlations that are found.
95-99: This sentence is awkward and hard to read.
The sentence has been shortened.
104: possible circular argument: Why exactly should poikilohydry relate to abundance? Could explain that non-frozen water availability constrains growth.
Sentence changed, we think this removes the problem,
110: In addition to growth and richness, this gradients section could benefit from mention of "local abundance" (frequency/cover/biomass) as a monitoring variable of interest; just a thought.
This is a good thought, indeed, unfortunately the data do not exist.
114: "Gradients will... show in which direction change will happen": Not necessarily true: e.g., warming may be greater (and faster) at a currently "cool" site than at a "warm" site; or vice-versa! Your statement doesn't exclude alternative outcomes.
The sentence has been modified but the overall concept that if it warms then changes will move in the direction indicated by the gradient is reasonable.
137: Is species number estimated at a single site? in a region? in a latitudinal band? Data collection methods are never really explained.
This is a reasonable question and the answer is in the reference that is now cited. The size of the areas will be different and they are for coastal areas around bases. An area effect is always possible but seems not to have a great impact on the numbers data. This is an Antarctic problem and there is no simple solution.
152-154, and 154-156: No evidence presented for these claims. Did someone simultaneously measure microsite water availability in relation to macroclimate? Does not appear that Green et al. (2011) explicitly did so.
This is an interpretation made in the cited paper and is based on in field experience and on the actual regression. Numbers of species at any site in the microenvironmental zone are highly variable because of water availability whereas the numbers are significantly related to precipitation in the more northerly macroenvironmental zone. This concept has been looked at by others during its generation.
The evidence is the regression.
172-173: I'd accept that carbon-balance physiological mechanisms could influence growth rates. But less clear is how physiology need necessarily influence species numbers (why can't one site have 50 poorly-performing species, and another have 5 excellently-performing species? or vice-versa?). There's no evidence to rule out either outcome.
The comment is based on the changes in allocation of carbon within the lichen. There is a change from allocation of survival to allocation to growth (reference given). Similarly, each species has its own allocation strategy and it appears that species cannot do everything but always have to make some sort of specialization choice. Hence, as suggested, change along a temperature gradient is mainly be change in species rather than a single species adapting. As it warms or water becomes more available then total productivity increases opening up new niches and new solutions (ie species) by the lichens.
178: Your glacial refugia hypothesis may be consistent with the present distribution, but further evidence would be needed to exclude alternative hypotheses. How would you rule out some other alternative (like post-LGM arrival from other continents and habitat filtering totally independent of glaciation)? What about population genetic structure, or a mechanistic dispersal model?
There is good evidence that cyanobacteria are inefficient below 0°C, both photosynthesis and nitrogen fixation are stopped. Free-living cyanobacteria go much further south, to 79°S, but only occur where there is free water at some time, ie where the temperature goes above 0°C. Lichens are not in water so most of the time are below freezing point. The physiological basis for this comment is good.
186-188: This sentence is awkward and hard to read.
rewritten
199: Usually we report r2 or R2 and a p-value when making a statement about correlation.
Information added
220-221: But surely competition is not negligible among saxicolous and terricolous lichen thalli... See work by Armstrong, Maestre, Pastore, ...
This is a newly exposed site in front of a retreating glacier and the rock surfaces are only partially colonized. So, actual thallus to thallus interactions are not likely.
222-227: Not sure why this reiterates lines 192-199, from references [33,34].
Most of this has been removed and only a brief lead-in remains.
242: "optimum temperature for photosynthesis" finally suggests how climate may impact the physiological basis of your observed growth changes. Perhaps elevate this sooner or more frequently throughout?
Actually it is not the main driver; the section has been rewritten to show this.
247: Your statement neglects that ecosystem net primary productivity necessarily includes respiration losses (from cellular respiration, or microbial decomposition) that might outpace photosynthetic gains -- what is your expectation for ecosystem respiration?
We write about net photosynthesis and its response to temperature; NP increases with rise in temperature below the optimal temperature so one would expect a greater carbon gain.
249: You present no evidence that biomass was measured or estimated -- perhaps you assumed that radial growth must yield more biomass? But that's not necessarily true if larger thalli are also thinner.
The section before this has been rewritten and now brings in the activity period which is the main driver. There are certainly many more processes that could be measured in Antarctica but the reality is that we are only at the beginning and, at most sites, only summer studies occur.
252-264: This is a very nice concise summary justifying your program.
266: Might be useful to mention that saxicolous lichens are somewhat decoupled from soil moisture (if indeed you continue to refer only to saxicolous), and so may be useful indicators of atmospheric process.
Some changes made to draw attention to this point
277-278: So departures from a pure-climate model, therefore, could indicate the magnitude of dispersal limitation (i.e., how far from equilibrium).
This point, which is very nice, has been added.
291: Not sure why this reiterates lines 172-173, which are likewise unsubstantiated (contradictory outcomes equally plausible/possible).
Section mainly removed
312-320: I don't doubt this may be true, but no evidence is given! This is probably consistent with the authors' detailed and extensive natural history observations at great effort, but the claim can only be exclusively supported with both microclimate/macroclimate measurements to the exclusion of alternative explanations.
See previous comments about this point, the suggestions are abased on the regression with temperature which fails in the southern regions where precipitation is also very low.
325-33: This is a concise, clear paragraph showing how physiology may scale to landscapes - nice!
347-353: Nice that you mention molecular tools, but might also help to mention that molecular studies can reveal photobiont switching which may expand/constrain climatic tolerances of the symbiosis.
This has now been mentioned with citations.
352: What is internal quarantine? Is this a biological concept of gene flow, or a social principle to prevent novel introductions?
This has been clarified, thanks
### end ###
Submission Date
08 February 2019
Date of this review
12 Feb 2019 16:11:56

Round 2
Reviewer 1 Report
I believe that the manuscript has improved substantially and its current version can be accepted for publication.
Author Response
Thank you for agreeing with our changes and answers. We feel that your review has substantially contributed to the improvement of the paper.
Reviewer 3 Report
Overall comments:
------------------------------------------------------------------------------------------
1. Many of my review comments were ignored or elided.
2. Claims continue to be inductive and speculative, rather than deductive and conclusive. Causal claims persist where only correlative ones are warranted.
3. Methods and data remain insufficiently described and still hidden deep within original literature.
------------------------------------------------------------------------------------------
Specifically (and referring to *original* line numbers):
10: You "accept" my comment but did not make appropriate changes. The word "saxicolous" only appears once in the whole paper! You need to clearly convey, at the outset and throughout, that inferences apply to rock-dwelling taxa.
72: My comment pertained to *abiotic* evidence (warming? drying? seasonality shifts?). Consider that climate impact = abiotic climate change + biological response, but you still only discuss biological responses.
79: We don't agree. Sure your correlation/regression lets you "predict" in a statistical sense, but not in the logical sense of having rejected sufficiently realistic alternatives using very basic hypothesis testing! My root concern is that you make lichen climate biology a "just so story" whereby explanations match evidence purely by happenstance (inductive), rather than by robust rejection of alternatives (deductive)! Perhaps a better word here is simply "anticipate".
110: You still fail to articulate how homoiohydry/poikilohydry should directly link to abundance. For example, does poikilohydry permit greater individual thallus growth under frozen-water/water-limiting conditions such as those found in Antarctica? Then how should individual growth scale to population abundance?
114: Your response fails to address my original comment. Agreed that your assumption is reasonable, but so are many alternative assumptions (such as that climatic warming may be anisotropic). This duality is the very definition of inconclusive, so why waste words?
137: Why do you continue to bury methods deep inside cited papers? This is opaque! A review should briefly summarize and weave together original sources, not force readers to do the hunting themselves!
152-156: Causal claims continue to be unsubstantiated. Perhaps just change "driven by" to "associated with" to distinguish causation from correlation. Your regression is a tool of correlation, nothing more, and does not forbid alternative explanations.
172: Clearly C-allocation strategies vary among species, and such physiological variation translates into the *identity* of species that may persist at any point along a gradient (i.e., turnover). But you still fail to articulate how C physiology is linked to the *number* of species at a site.
178: Your response fails to address my original comment: observed cyano distributions are consistent with your Antarctic glacial refugia hypothesis, but how would you rule out the alternative hypothesis of post-LGM immigration from other continents? Without doing so (probably via phylogeographic analysis), you are telling a "just-so story". Should just abandon the claim then.
220: Your response states why you assume that interactions are negligible, but rather you should tell the reader that within the manuscript!
247: Your response fails to address my original comment: what if higher temperature fostered higher microbial decomposition that negates any lichen growth gains (and so tundra Net Primary Productivity declines)? Without knowledge of decomposition, you have no evidence whatsoever to make claims about ecosystem productivity.
249: Your response fails to address my original comment: why do you make a biomass claim when biomass was not measured?
### end ##
Author Response
Answer to review 3:
10: You "accept" my comment but did not make appropriate changes. The word "saxicolous" only appears once in the whole paper! You need to clearly convey, at the outset and throughout, that inferences apply to rock-dwelling taxa.
Sorry, we overlooked these changes. This has been noted in 4 places, including Abstract (lines 13, 167, 227 and 272). .
72: My comment pertained to *abiotic* evidence (warming? drying? seasonality shifts?). Consider that climate impact = abiotic climate change + biological response, but you still only discuss biological responses.
We only look at biological change as the aim of the article is to draw attention to the great potential for such measurements in Antarctica. We also draw attention to the importance of the active period which is an effective integrator of abiotic effects.
79: We don't agree. Sure your correlation/regression lets you "predict" in a statistical sense, but not in the logical sense of having rejected sufficiently realistic alternatives using very basic hypothesis testing! My root concern is that you make lichen climate biology a "just so story" whereby explanations match evidence purely by happenstance (inductive), rather than by robust rejection of alternatives (deductive)! Perhaps a better word here is simply "anticipate".
The reviewer makes a good point, however, this is rather a conceptual discussion. The first role of scientists is to observe, and observation leads to conclusions about links between organisms and their environments, correlations. Correlations only indicate a link, however, when explanations for such a link are available then it is appropriate to use them. In this case we offer physiological evidences to explain lichen response to warming and cooling. This is not just an “inductive” statement. Any way we might accept using “anticipate” instate of predict (line 81).
110: You still fail to articulate how homoiohydry/poikilohydry should directly link to abundance. For example, does poikilohydry permit greater individual thallus growth under frozen-water/water-limiting conditions such as those found in Antarctica? Then how should individual growth scale to population abundance?
We are not speaking about abundance anymore. We are using “dominance” instead. The simple fact that higher plants are excluded from the vast majority of Antarctica is a good indicator of the ecological limits of these organisms. Actual depth of cold is not the limit or, it appears to be a requirement for a certain level of warmth during the year to allow completion of metabolic processes.
114: Your response fails to address my original comment. Agreed that your assumption is reasonable, but so are many alternative assumptions (such as that climatic warming may be anisotropic). This duality is the very definition of inconclusive, so why waste words?
We still think that gradients indicate the direction of changes and what can be expected. This is exactly the same as in mountain altitudinal gradients. Local variation away from the main gradient trends is a reasonable and represents microclimate effects. In Arctic studies the abundance also follows local microclimate so would follow temperature rather than latitude.
137: Why do you continue to bury methods deep inside cited papers? This is opaque! A review should briefly summarize and weave together original sources, not force readers to do the hunting themselves!
Here we summarize the published results from floristic papers affecting different areas and the methods used, which are usually standard methods, are available in these papers. We do not discuss these in detail as they all used standard methodologies which are fully described in the papers.
152-156: Causal claims continue to be unsubstantiated. Perhaps just change "driven by" to "associated with" to distinguish causation from correlation. Your regression is a tool of correlation, nothing more, and does not forbid alternative explanations.
We have checked all uses of the word driver and replaced it in some cases. In others it is used as a general term and mainly is the role of suggesting future possibilities. We agree with the limitations involved with using correlations, correlations do not suggest actual direct linkages but, and it is an important but, where potential mechanisms are known it is acceptable to drew attention to the links.
172: Clearly C-allocation strategies vary among species, and such physiological variation translates into the *identity* of species that may persist at any point along a gradient (i.e., turnover). But you still fail to articulate how C physiology is linked to the *number* of species at a site.
There is a general law that links productivity with diversity along natural ecosystems, independently on the identity of the species. Each species represents a solution to survival at a given point and allocation of resources by an individual species is how such solutions are achieved. Most publications overlook allocation as it has been little studied. One aim of this paper is to stress that it exists and that Antarctica is potentially an excellent area for such studies because of the large environmental gradients.
178: Your response fails to address my original comment: observed cyano distributions are consistent with your Antarctic glacial refugia hypothesis, but how would you rule out the alternative hypothesis of post-LGM immigration from other continents? Without doing so (probably via phylogeographic analysis), you are telling a "just-so story". Should just abandon the claim then.
Potential refugia are available, the most likely being offshore islands which would have had a milder climate. We have made the statement more general and also added a reference (lines 186-187).
220: Your response states why you assume that interactions are negligible, but rather you should tell the reader that within the manuscript!
Curiously this point seems to have ended up only in the Abstract, it has now been included in the text in the lichen growth rates section.
247: Your response fails to address my original comment: what if higher temperature fostered higher microbial decomposition that negates any lichen growth gains (and so tundra Net Primary Productivity declines)? Without knowledge of decomposition, you have no evidence whatsoever to make claims about ecosystem productivity.
It should not be forgotten that we are speaking about saxicolous lichens that are complete individuals and not dramatically affected by microbial decomposition when growing. The main point is that, although maximal rates of net photosynthesis appear to be relatively stable across Antarctica the actual temperatures when active are always below the optimal temperature for net photosynthesis. Thus, any increase in temperature will cause an increase in productivity (it is net photosynthesis so respiratory costs are within the measurement).
249: Your response fails to address my original comment: why do you make a biomass claim when biomass was not measured?
Actually, we assume that radial growth must yield more biomass because crustose lichens like Rhizocarpon or Buellia (and most of crustose lichens) do not get thinner when larger. There is also a gradient in vegetation cover across Antarctica with much higher cover values in the warmer areas,
### end ##
Answer to review 3:
10: You "accept" my comment but did not make appropriate changes. The word "saxicolous" only appears once in the whole paper! You need to clearly convey, at the outset and throughout, that inferences apply to rock-dwelling taxa.
Sorry, we overlooked these changes. This has been noted in 4 places, including
Abstract (lines 13, 167, 227 and 272).
72: My comment pertained to *abiotic* evidence (warming? drying? seasonality
shifts?). Consider that climate impact = abiotic climate change +
biological response, but you still only discuss biological responses.
We only look at biological change as the aim of the article is to draw attention to the great potential for such measurements in Antarctica. We also draw attention to the importance of the active period which is an effective integrator of abiotic effects.
79: We don't agree. Sure your correlation/regression lets you "predict" in a statistical sense, but not in the logical sense of having rejected sufficiently realistic alternatives using very basic hypothesis testing! My root concern is that you make lichen climate biology a "just so story" whereby explanations match evidence purely by happenstance (inductive), rather than by robust rejection of alternatives (deductive)! Perhaps a better word here is simply "anticipate".
The reviewer makes a good point, however, this is rather a conceptual discussion. The first role of scientists is to observe, and observation leads to conclusions about links between organisms and their environments, correlations. Correlations only indicate a link, however, when explanations for such a link are available then it is appropriate to use them. In this case we offer physiological evidences to explain lichen response to warming and cooling. This is not just an “inductive” statement. Any way we might accept using “anticipate” instate of predict (line 81).
110: You still fail to articulate how homoiohydry/poikilohydry should directly link to abundance. For example, does poikilohydry permit greater individual thallus growth under frozen-water/water-limiting conditions such as those found in Antarctica? Then how should individual growth scale to population abundance?
We are not speaking about abundance anymore. We are using “dominance” instead. The simple fact that higher plants are excluded from the vast majority of Antarctica is a good indicator of the ecological limits of these organisms. Actual depth of cold is not the limit or, it appears to be a requirement for a certain level of warmth during the year to allow completion of metabolic processes.
114: Your response fails to address my original comment. Agreed that your assumption is reasonable, but so are many alternative assumptions (such as that climatic warming may be anisotropic). This duality is the very definition of inconclusive, so why waste words?
We still think that gradients indicate the direction of changes and what can be expected. This is exactly the same as in mountain altitudinal gradients. Local variation away from the main gradient trends is a reasonable and represents microclimate effects. In Arctic studies the abundance also follows local microclimate so would follow temperature rather than latitude.
137: Why do you continue to bury methods deep inside cited papers? This is opaque! A review should briefly summarize and weave together original sources, not force readers to do the hunting themselves!
Here we summarize the published results from floristic papers affecting different areas and the methods used, which are usually standard methods, are available in these papers. We do not discuss these in detail as they all used standard methodologies which are fully described in the papers.
152-156: Causal claims continue to be unsubstantiated. Perhaps just change "driven by" to "associated with" to distinguish causation from correlation. Your regression is a tool of correlation, nothing more, and does not forbid alternative explanations.
We have checked all uses of the word driver and replaced it in some cases. In others it is used as a general term and mainly is the role of suggesting future possibilities. We agree with the limitations involved with using correlations, correlations do not suggest actual direct linkages but, and it is an important but, where potential mechanisms are known it is acceptable to drew attention to the links.
172: Clearly C-allocation strategies vary among species, and such physiological variation translates into the *identity* of species that may persist at any point along a gradient (i.e., turnover). But you still fail to articulate how C physiology is linked to the *number* of species at a site.
There is a general law that links productivity with diversity along natural ecosystems, independently on the identity of the species. Each species represents a solution to survival at a given point and allocation of resources by an individual species is how such solutions are achieved. Most publications overlook allocation as it has been little studied. One aim of this paper is to stress that it exists and that Antarctica is potentially an excellent area for such studies because of the large environmental gradients.
178: Your response fails to address my original comment: observed cyano distributions are consistent with your Antarctic glacial refugia hypothesis, but how would you rule out the alternative hypothesis of post-LGM immigration from other continents? Without doing so (probably via phylogeographic analysis), you are telling a "just-so story". Should just abandon the claim then.
Potential refugia are available, the most likely being offshore islands which would have had a milder climate. We have made the statement more general and also added a reference (lines 186-187).
220: Your response states why you assume that interactions are negligible, but
rather you should tell the reader that within the manuscript!
Curiously this point seems to have ended up only in the Abstract, it has now been included in the text in the lichen growth rates section.
247: Your response fails to address my original comment: what if higher
temperature fostered higher microbial decomposition that negates any lichen
growth gains (and so tundra Net Primary Productivity declines)? Without
knowledge of decomposition, you have no evidence whatsoever to make claims
about ecosystem productivity.
It should not be forgotten that we are speaking about saxicolous lichens that are complete individuals and not dramatically affected by microbial decomposition when growing. The main point is that, although maximal rates of net photosynthesis appear to be relatively stable across Antarctica the actual temperatures when active are always below the optimal temperature for net photosynthesis. Thus, any increase in temperature will cause an increase in productivity (it is net photosynthesis so respiratory costs are within the measurement).
249: Your response fails to address my original comment: why do you make a biomass claim when biomass was not measured?
Actually, we assume that radial growth must yield more biomass because
crustose lichens like Rhizocarpon or Buellia (and most of crustose lichens) do
not get thinner when larger. There is also a gradient in vegetation cover
across Antarctica with much higher cover values in the warmer areas,
### end ##
